# Physico-Chemical Modifications Affecting the Activity and Stability of Cu-Based Hybrid Catalysts during the Direct Hydrogenation of Carbon Dioxide into Dimethyl-Ether

**DOI:** 10.3390/ma15217774

**Published:** 2022-11-04

**Authors:** Fabio Salomone, Giuseppe Bonura, Francesco Frusteri, Micaela Castellino, Marco Fontana, Angelica Monica Chiodoni, Nunzio Russo, Raffaele Pirone, Samir Bensaid

**Affiliations:** 1Department of Applied Science and Technology (DISAT), Politecnico di Torino, Corso Duca degli Abruzzi 24, 10129 Turin, Italy; 2Consiglio Nazionale delle Ricerche-Istituto di Tecnologie Avanzate per l’Energia “Nicola Giordano” (CNR-ITAE), Via Santa Lucia Sopra Contesse 5, 98126 Messina, Italy; 3Istituto Italiano di Tecnologia (IIT), Via Livorno 60, 10144 Turin, Italy

**Keywords:** CO_2_ hydrogenation, hybrid catalysts, zeolites, ferrierite, DME, methanol

## Abstract

The direct hydrogenation of CO_2_ into dimethyl-ether (DME) has been studied in the presence of ferrierite-based CuZnZr hybrid catalysts. The samples were synthetized with three different techniques and two oxides/zeolite mass ratios. All the samples (calcined and spent) were properly characterized with different physico-chemical techniques for determining the textural and morphological nature of the catalytic surface. The experimental campaign was carried out in a fixed bed reactor at 2.5 MPa and stoichiometric H_2_/CO_2_ molar ratio, by varying both the reaction temperature (200–300 °C) and the spatial velocity (6.7–20.0 NL∙g_cat_^−1^∙h^−1^). Activity tests evidenced a superior activity of catalysts at a higher oxides/zeolite weight ratio, with a maximum DME yield as high as 4.5% (58.9 mg_DME_∙g_cat_^−1^∙h^−1^) exhibited by the sample prepared by gel-oxalate coprecipitation. At lower oxide/zeolite mass ratios, the catalysts prepared by impregnation and coprecipitation exhibited comparable DME productivity, whereas the physically mixed sample showed a high activity in CO_2_ hydrogenation but a low selectivity toward methanol and DME, ascribed to a minor synergy between the metal-oxide sites and the acid sites of the zeolite. Durability tests highlighted a progressive loss in activity with time on stream, mainly associated to the detrimental modifications under the adopted experimental conditions.

## 1. Introduction

Nowadays, the increase in global warming is a crucial issue which has to be coped with. The development of catalysts and processes for producing synthetic fuels as an alternative to traditional fossil resources has indeed recently received particular attention in the power-to-fuels research field [1,2,3,4,5,6,7,8]. Among several technologies, power-to-fuel seems to be one of the most promising to produce a range of synthetic fuels and chemicals (e.g., methane, methanol, dimethyl-ether, light and heavy hydrocarbons, etc.) for the purpose of decarbonizing society and carbon footprint reduction [4,9,10,11,12,13,14]. The great advantage of this technology is the exploitation of waste CO_2_ that could be captured and recovered by means of carbon capture and utilization (CCU) systems [4,7,8,9,15]. On the other hand, the CO_2_ could be converted in useful value-added carbon-based products by using renewable H_2_ that could be produced with many technologies (e.g., electrolysis, biomass valorization, aqueous phase reforming, etc.) [16,17,18,19,20,21] by exploiting different sources of renewable energy (e.g., photovoltaic, wind power, geothermal, hydroelectric, etc.). Among those power-to-fuel processes, the synthesis of dimethyl-ether (DME) via CO_2_ hydrogenation seems to be a promising process to obtain a useful, valuable and environmentally friendly product. DME is indeed an added-value product with a high cetane rating and low pollutant emissions during the combustion; thus, it could be used as a substitute of diesel fuel. Moreover, DME is a key intermediate for producing chemicals or petrochemicals [1,13,22,23,24,25,26].

Industrially, DME is generally produced from syngas through the most mature pathway, which involves a two-step process: the synthesis of methanol (MeOH) on a Cu-based catalyst, which is economically cheaper than noble metals, and a subsequent dehydration of MeOH to DME on an acid catalyst [22,24,25,27]. This process is strongly limited by thermodynamics and the common operating condition ranges between 260–280 °C and 5–10 MPa because at a low temperature the CO_2_ conversion is kinetically limited and at high temperatures prevails the reverse water gas shift reaction [10,24,27,28,29]. Considering the attention paid to CCU technologies, many researchers have recently shown a growth of interest in CO_2_ hydrogenation. In more detail, there are several technologies to convert CO_2_ and H_2_ into methanol: thermo-catalysis, electro-catalysis, photo-catalysis, plasma-assisted catalysis and even biological processes [10,30,31,32,33]. This work focuses on the thermocatalytic route and to overcome the thermodynamic issues of the methanol synthesis via CO_2_ hydrogenation, a single-step process has been investigated. There are two ways that could be investigated: (i) the reactor configuration and (ii) the catalyst design. On the one hand, membrane reactors could be used to selectively remove the products favoring a shift of the reaction toward methanol and DME [34,35]. On the other hand, a bifunctional catalyst that could couple the CO_2_ conversion into methanol and its subsequent dehydration into dimethyl-ether could be another solution to overcome the thermodynamic limitations of the methanol synthesis [22,36,37,38,39,40,41,42,43,44]. Therefore, the design of novel hybrid catalysts has been investigated for the purpose of directly producing DME in a single-step process [24,25,45,46,47,48]. The net reaction of DME synthesis is reported in Equation (1); however, the overall process includes a series of reactions: the CO_2_ hydrogenation to methanol (2), the reverse water gas shift (RWGS) reaction (3), the CO hydrogenation to methanol (4) and the methanol dehydration to DME (5). Interestingly, in the single step process the thermodynamic limitations of the methanol synthesis could be overcome due to its consecutive dehydration into DME; hence, the CO_2_ per pass conversion could be increased even at low temperatures [9,24,27].
(1)$$2\;{\mathrm{CO}}_{2(g)}{+6\;H}_{2(g)}\;⇄{\;\mathrm{CH}}_{3}{\mathrm{OCH}}_{3(g)}{+3\;H}_{2}O_{\left(g\right)}\;\Delta {\tilde{H}}_{298.15}^{0}=-122.2\;kJ/mol$$
(2)$${\mathrm{CO}}_{2(g)}{+3\;H}_{2(g)}\;⇄{\;\mathrm{CH}}_{3}{\mathrm{OH}}_{\left(g\right)}{+H}_{2}O_{\left(g\right)}\;\Delta {\tilde{H}}_{298.15}^{0}=-49.4\;kJ/mol$$
(3)$${\mathrm{CO}}_{2(g)}{+H}_{2(g)}\;⇄{\mathrm{CO}}_{\left(g\right)}{+H}_{2}O_{\left(g\right)}\;\Delta {\tilde{H}}_{298.15}^{0}=+41.2\;kJ/mol$$
(4)$${\mathrm{CO}}_{\left(g\right)}{+2\;H}_{2(g)}\;⇄{\;\mathrm{CH}}_{3}{\mathrm{OH}}_{\left(g\right)}\;\Delta {\tilde{H}}_{298.15}^{0}=-90.4\;kJ/mol$$
(5)$${2\;\mathrm{CH}}_{3}{\mathrm{OH}}_{\left(g\right)}⇄{\;\mathrm{CH}}_{3}{\mathrm{OCH}}_{3(g)}{+H}_{2}O_{\left(g\right)}\;\Delta {\tilde{H}}_{298.15}^{0}=-23.4\;kJ/mol$$

For methanol synthesis, the most common Cu/ZnO-based formulations are typically completed by a ceramic material, for instance alumina (Al_2_O_3_), zirconia (ZrO_2_) or silica (SiO_2_) in order to increase the surface area and to prevent sintering of Cu particles [9,24,33,49,50,51,52]. These catalysts are remarkably selective toward methanol, while the main by-product is CO due to the endothermic RWGS reaction [9,27]. Existing research has demonstrated that CO_2_ hydrogenation to methanol is favored over Cu/ZnO-based catalysts, which reduces the activation energy of the reaction, whereas the activation energy of CO hydrogenation over Cu/ZnO-based catalysts is greater than over a Cu-based catalyst [33,53,54,55].

Concerning the acid catalyst, the performance of many materials has been explored in the open literature. Zeolites (for instance MFI, Theta-1, mordenite and ferrierite) and γ-alumina are the most investigated catalysts for dehydrating methanol to DME [9,40,45,46,47,48,56,57]. Frusteri et al. (2017) have recently revealed that ferrierite is one of the most promising acid catalysts for this purpose, compared to mordenite and MFI [40]. The number, the distribution and the strength of the acid sites are the most important features of these catalysts [56,58]. Generally, an increase in acid sites, uniformly distributed on the surface of the material, causes an increase in DME yield. Furthermore, some authors have shown that weak and medium strong acid sites are active in methanol dehydration; however, some correlation between the increase in DME yield and the higher concentration of strong acid sites have been also reported [22,25,40,56,58]. Recently, Catizzone et al. (2021) demonstrated that nano-sized zeolites are more performant than the micro-sized in methanol dehydration to DME by increasing the DME selectivity and reducing the carbon deposition [59]; and this effect of the crystallinity has also been observed in a previous work of Bonura et al. (2018) [56].

Really, many synthesis techniques have been explored in order to obtain hybrid catalysts for direct CO_2_ hydrogenation to DME. The most common techniques are physical mixtures, coprecipitation and wet impregnation [9,22,40]. Each technique has many issues that must be considered, for instance, the control of synthesis conditions and their repeatability, the composition and the distribution of the metallic loading on the surface of the acid catalyst, the dimensions of the particles and their homogeneity. In more detail, the gel-oxalate co-precipitation technique has been investigated in the literature and researchers have observed that it led to the formation of ultra-fine oxides particles in which the zinc is incorporated in the structure of the oxalic copper [60,61,62]. After calcination and reduction in H_2_, the catalytic structure exhibited a high activity in CO_2_ hydrogenation and a high methanol selectivity [60,61,62]. Many authors have also reported the need for a high metal dispersion for facilitating CO_2_ activation, while many others reported as more crucial the extent of metal-oxide interface for an effective CO_2_ activation [13,22,23,25,40,56,57]. However, few studies have investigated the deactivation phenomena that involve the catalysts during on-stream operating conditions [13,45,51,57,63].

Therefore, this work aims to contribute to this growing area of research by exploring the catalytic performance of ferrierite-based CuZnZr catalysts during the direct CO_2_ hydrogenation to DME, paying attention to the influence both of preparation and composition on activity, selectivity and stability at different on-stream reaction conditions. On this account, several characterization techniques were used in order to individuate peculiar structure–activity relationships and significant modifications between calcined and spent catalysts.

## 2. Materials and Methods

### 2.1. Preparation of the Catalytic System

The procedure adopted for the preparation of the CuZnZr-based hybrid systems consists of a combination of copper, zinc and zirconia, which are commonly used for methanol synthesis, and a commercial ferrierite-type zeolite as described in our previous works [22,40,64]. Four different CuZnZr-based catalysts for DME synthesis were prepared. Two catalysts were synthesized via gel-oxalate coprecipitation following the procedure elsewhere reported [22,40]: copper, zinc and zirconyl nitrates were solubilized in ethanol (Cu/Zn/Zr atomic ratio 6:3:1) and then coprecipitated by oxalic acid at ambient conditions under vigorous stirring in a solution containing finely dispersed ferrierite powder [22,40]. After filtration, drying (95 °C for 16 h) and calcination (350 °C for 4 h), the two samples were designated as CZZ/FER OX 1:2 and CZZ/FER OX 2:1, being characterized by an oxides/zeolite weight ratio of 1:2 and 2:1, respectively. The third catalyst was prepared by the wet impregnation technique: the ferrierite powder was impregnated by the metal precursors solubilized in ethanol (Cu/Zn/Zr atomic ratio 6:3:1), dried and then calcined [22]. This sample was named CZZ/FER WI 1:2 (CZZ/FER weight ratio of 1:2). Lastly, the fourth sample denoted as CZZ-FER MIX 1:2 was obtained as a physical mixture of coprecipitated CuZnZr (gel-oxalate coprecipitation, Cu/Zn/Zr atomic ratio 6:3:1) and ferrierite powder with a CuZnZr/FER weight ratio of 1:2.

Each catalyst powder was pelletized by applying 10.6 kN (=15 MPa), ground in an agate mortar and sieved for obtaining small particles (particle size range: 250–500 μm).

As a reference, a ternary CuO-ZnO-ZrO_2_ catalyst for CO_2_ hydrogenation to methanol was prepared by gel-oxalate co-precipitation with a Cu/Zn/Zr atomic ratio 6:3:1.

### 2.2. Characterization of the Catalysts

The textural properties of both calcined and spent catalysts were determined by N_2_ physisorption at −196 °C using a Micromeritics Tristar II ASAP 3020 analyzer. The samples (~80 mg) were previously outgassed at 300 °C in inert (N_2_) gas for 2 h using a Micromeritics FlowPrep 060 to remove water and other atmospheric contaminants from the surface. The total surface area was calculated by means of Langmuir’s theory (the 0.05 ≤ p/p° ≤ 0.35 relative pressure range was used) [65] because the Brunauer–Emmett–Teller (BET) method [66] could not be applied due to the negative value of the BET constant for the ferrierite-based catalysts. Moreover, the surface area and volume of micropores were evaluated via the t-plot method using the Harkins and Jura equation for the estimation of the statistical thickness [67,68,69], while the porosimetry analysis of meso- and macropores was performed by applying the Barrett–Joyner–Halenda (BJH) algorithm to the desorption branch of the isotherm [70,71].

The powders of both calcined and spent catalysts were investigated in a Philips X’Pert PW3040 diffractometer operating at 40 kV and 40 mA, using a Ni β-filtered Cu-Kα radiation characterized by a wavelength (λ) equal to 1.5406∙10^−10^ m. X-ray diffraction (XRD) patterns were collected at room temperature over the 2θ angle range of 5–90°, with a scan step size of 0.013° 2θ and a time per step equal to 0.3 s, while the average crystallite size (d_c_, nm) was estimated according to Scherrer’s equation [72].

Both calcined and spent catalysts were observed with a field emission scanning electron microscope (FE-SEM) Zeiss Merlin equipped with a Gemini-II column for the purpose of analyzing their morphology. Moreover, energy dispersive X-ray spectroscopy (EDS) was employed to determine the elemental composition of portions of the catalysts. Furthermore, both morphology and crystalline structure of the samples were investigated by a transmission electron microscopy (TEM) with FEI Tecnai F20 ST microscope operating at 200 kV acceleration voltage. Concerning sample preparation, each sample powder was dispersed in ethanol (purity > 99.8%) through sonication for 2 min and subsequently drop-casted on a Cu holey-carbon TEM grid.

For the elemental composition of the catalysts, an inductively coupled plasma mass spectrometer (iCAP Q ICP-MS, Thermo Fisher Scientific, Waltham, MA, USA) was employed. Approximately 100 mg of each catalyst was digested in 10 mL of acid solution (6 mL of HCl 37 vol.%, 2 mL of HNO_3_ 65 vol.% and 2 mL of HF 48 vol.%) by using a Milestone ETHOS EASY SK15. The solution was heated at 10 °C/min from room temperature to 220 °C and held for 15 min at the maximum temperature, then it was cooled at room temperature in approximately 30 min. Each concentrated solution was diluted using milli-q water to have the concentration of all elements within the calibration range (100–2000 ppb), then these samples were analyzed using ICP-MS.

TPR measurements were performed in a Thermoquest TPD/R/O 1100 analyzer, equipped with a thermal conductivity detector (TCD). A mass of 20 mg of calcined catalyst was placed in a quartz tube reactor between two layers of quartz wool. To ensure the complete oxidation of metals, each sample was pre-treated in pure O_2_ flow (40 mL/min), heating the oven from room temperature to the calcination temperature (350 °C) [22] with a constant heating rate of 10 °C/min and holding it for 30 min. Subsequently the sample was cleaned in He flow (40 mL/min) at 350 °C for 30 min and then it was cooled to 40 °C. The H_2_-TPR was carried out under a constant flow (20 mL/min) of 5.000 vol.% H_2_/Ar, heating the sample from 40 °C to 900 °C with a heating rate of 10 °C/min.

The surface of the catalysts was investigated via X-ray photoelectron spectroscopy (XPS) employing a PHI 5000 Versa Probe equipment, using a band-pass energy of 187.85 eV, a take-off angle of 45° and an X-ray spot diameter of 100 μm; high resolution spectra were collected using a band-pass energy of 23.50 eV.

Surface concentration of acid sites was evaluated by means of ammonia (NH_3_) pulse chemisorption in the Thermoquest TPD/R/O 1100 analyzer. The analyses were performed placing 90 mg of catalyst in a quartz tube reactor. Each sample was pre-treated flowing He (25 mL/min) from room temperature to 120 °C, then it was reduced using 5 vol.% H_2_/Ar (25 mL/min), heating the sample up to 350 °C with a heating rate of 10 °C/min and holding it for 60 min at the maximum temperature. After that, the sample was cleaned with He (25 mL/min) for 30 min at 350 °C and then cooled down to 100 °C in inert flow. The NH_3_ pulse chemisorption analysis was carried out in isothermal conditions (100 °C) using 9.99 vol.% NH_3_/He and He as carrier gas (25 mL/min).

### 2.3. Catalytic Tests

The catalytic activity of each sample (catalyst load = 1.5 g) was investigated in a fixed bed stainless steel reactor. The catalytic bed had an annular section (i.d., 3 mm; o.d., 8 mm) due to the presence of an innertube in which a thermocouple was inserted for the measurement of the reaction temperature within the catalytic bed. The catalytic bed volume varied depending on the apparent density of each catalyst (as shown in Appendix A); therefore, to compare the results, all the catalytic tests were carried out keeping constant the catalyst load while varying the weight hourly space velocity (WHSV, NL∙g_cat_^−1^∙h^−1^). Each sample was pre-treated by using 30 NL/h (10 vol.% H_2_/N_2_) at 0.2 MPa and 350 °C for 3 h. Afterwards, a stability test was performed on each sample (20 h at 275 °C, 2.5 MPa and 13.3 NL∙g_cat_^−1^∙h^−1^; inlet composition: 60 vol.% H_2_, 20 vol.% CO_2_ and 20 vol.% N_2_). Activity tests were carried out on the hybrid catalysts at 2.5 MPa (inlet composition: 60 vol.% H_2_, 20 vol.% CO_2_ and 20 vol.% N_2_) both varying the reaction temperature between 200 °C and 300 °C with the WHSV between 6.7 NL∙g_cat_^−1^∙h^−1^ and 20 NL∙g_cat_^−1^∙h^−1^.

The gases at the reactor outlet were split into two streams. On the one side, a portion of the gases was directly analyzed using a gas chromatograph (7890B GC System, Agilent Technologies, Santa Clara, CA, USA) equipped with a heated transfer line (120 °C, atmospheric pressure), a two-column separation system (HP-PLOT/Q and HP-PLOT Molesieve) connected to a thermal conductivity detector (TCD) and a flame ionization detector (FID). On the other side, water and methanol were condensed in a tank (room temperature and 2.5 MPa), then the gaseous stream was completely dehydrated using a silica gel trap (room temperature and atmospheric pressure) and analyzed with an in-line X-STREAM EMERSON gas analyzer equipped with two nondispersive infrared (NDIR) sensors and a thermal conductivity detector (TCD) for monitoring CO, CO_2_ and H_2_ concentrations, respectively.

## 3. Results and Discussion

### 3.1. Physico-Chemical Characterization of the Catalysts

#### 3.1.1. Analytical Composition, Texture, Structure and Morphology

In Appendix A, the relative atomic concentration of each investigated sample is reported, as determined by ICP-MS characterization. Despite a slight discrepancy between the actual and nominal composition, the data reported suggest the effectiveness of the adopted preparation procedures and its suitability to achieve a very good control of the metal loading on all the samples.

The textural properties were investigated on both calcined and spent catalysts for the purpose of examining variations in terms of surface area, porosimetry and catalytic performance. As presented in Appendix A, the bare ferrierite exhibits a high specific surface area that matches the nominal value (400 m^2^/g_cat_), while the porosimetry analysis confirms the presence of a microporous structure and a broadened distribution of macropores, as can be seen in Appendix A [58,73]. Concerning the hybrid catalysts, their textural properties appear to be directly linked to the CuZnZr/ferrierite weight ratio. Nevertheless, the specific surface area of both calcined CZZ/FER OX 1:2 (311 m^2^/g_cat_) and CZZ/FER WI 1:2 (302 m^2^/g_cat_) is lower than the surface exposure of the CZZ-FER MIX 1:2 sample (337 m^2^/g_cat_), as a direct consequence of the different preparation method. In fact, the methanol synthesis phase which is deposited on the surface of the bare ferrierite (by coprecipitation or impregnation) reduces the sample microporosity. However, this loss in surface area is partially balanced by the mesoporosity of the oxides, as can be seen in Appendix A. More specifically, the coprecipitated catalysts show a sharpened mesopore size distribution at approximately 15 nm, whereas the spent catalysts exhibit a decrease in both the total specific surface area and the total pore volume. In addition, as illustrated in Figure 1, there is a direct correlation between the specific surface area and the micropore volume. Indeed, the slope of the straight line of the spent catalysts is greater than that of the calcined samples, suggesting a possible agglomeration of metal-oxides which leads to the occlusion of the ferrierite micropores and accordingly to a considerable decrease of the surface area [74,75].

X-ray diffractograms were collected for both calcined (see Appendix A) and spent (see Appendix A) catalysts to evidence the influence of the experimental conditions on the samples. The XRD pattern of calcined bare ferrierite (FER) is characterized by several intense and narrow peaks between 7° and 30° (PDF 00-044-0104 and PDF 01-088-1796), as also reported in the literature [22,76,77,78]. Its crystalline structure was not observable by TEM due to its instability under the beam and its instantaneous transformation from crystalline to amorphous structure [79]. The calcined CuZnZr-based hybrid catalysts (see Appendix A) show various reflections between 30° and 40°, corresponding to at least seven convoluted peaks belonging to two main crystalline phases: (i) the monoclinic phase of CuO (PDF 00-045-0937) at 32.5°, 35.5°, 38.7° and 38.9°; and (ii) the hexagonal structure of ZnO (PDF 00-036-1451) at 31.8°, 34.4° and 36.3°. The broadening of these peaks is principally caused by the small crystallite size of those phases (see Appendix A). Zirconia (ZrO_2_) was not detected due to its small amount; in addition, it is probably present as an amorphous phase. Data in Appendix A point out that CuO crystallite size is lower than 10 nm; the only exception is related to CZZ/FER WI 1:2 in which CuO crystallite size is approximately 18 nm. This fact is certainly linked to the synthesis technique in which the precursors form large clusters of precipitate during the evaporation of the solvent. As for the XRD patterns of spent catalysts (see Appendix A), three main reflection peaks of the cubic structure of metallic copper are clearly visible at 43.3°, 50.4° and 74.1° (PDF 00-004-0836). Those patterns reveal that CuO is not completely reduced to metallic Cu either in the ferrierite-free CuZnZr sample or in CZZ/FER OX 1:2 and CZZ-FER MIX 1:2 catalysts. Regarding the formation of Cu_2_O, its main characteristic peak at 36.4° overlaps with the most intense peak of ZnO located at 36.3°. It is noteworthy that the Cu crystallite size is larger than the CuO crystallite size for all the spent catalysts. Nevertheless, the coprecipitated hybrid catalysts have revealed a smaller increase in Cu crystallite size compared with the other spent catalysts. Differently, zinc oxide has a more stable structure and its crystallite size did not expand dramatically. Furthermore, considering a reduction temperature as high as 600 °C [80], ZnO is expected to remain in its oxidized form during the catalytic runs.

FESEM and EDS measurements were carried out on the samples to better understand their morphology and the metallic distribution on the zeolite. As shown in Figure 2, commercial FER, CZZ/FER OX 1:2 and CZZ/FER WI 1:2 were characterized by particles of ferrierite ranging approximately from 30 nm to 500 nm with different shapes.

Moreover, CZZ/FER OX 2:1 (see Figure 2d,e) revealed a completely different structure of the ferrierite-type zeolite in some portion of the catalyst; in fact, exposed a lamellar structure on which metals are deposited. Considering the relative oxides/zeolite weight ratio, CZZ/FER OX 1:2 has a low amount of metals dispersed on the surface of the zeolite. These metals do not completely cover the surface but form some sponge-like structures, which are characterized by small nanometric monocrystalline grains. However, the backscattered image and the EDS mapping of this catalyst (see Appendix A) illustrate a macroscopic uniformity of the metal load on the surface without evident inhomogeneities. Under the same preparation method, on CZZ/FER OX 2:1 a completely different disposition of the metals can be observed in FESEM images, evidenced by small aggregates of nanoparticles and a uniform coating on the zeolite surface. Similarly, CZZ/FER WI 1:2 exhibited an inhomogeneous deposition of the metallic elements on the external surface of the ferrierite with a typical layered structure (Figure 2g,h) [22], also confirmed by the backscattered FESEM image and the EDS mapping (see Appendix A) of the surface of this catalyst. A completely different macrostructure was observed for CZZ-FER MIX 1:2 catalyst, some portions of the sample being constituted by ferrierite and other parts by metals, as shown both in the backscattered FESEM image and in Appendix A. In general, all the samples exhibited a small size of metallic nanoparticles, accounting for a high metal dispersion which favors a synergistic effect among the active sites [40,55]. FESEM images do not have high resolution due to the low conductivity of the samples; hence, no macroscopic changes were observed between calcined and spent catalysts.

Bright-field TEM imaging provides an efficient way to visualize the distribution of the metal oxide nanoparticles with respect to the ferrierite structure due to difference in morphology and average atomic number. Figure 3 shows a comparison between representative low-magnification images of CuZnZr-based samples obtained with different preparation methods (gel-oxalate co-precipitation, wet impregnation, physical mixing). It is interesting to notice that a higher oxide loading (such as in the CZZ/FER OX 2:1 sample) results in a more homogenous distribution of the metal oxide nanoparticles on ferrierite. Regardless of the loading, the co-precipitation approach leads to a better inter-dispersion between metal oxide nanoparticles and ferrierite, if compared to wet impregnation (CZZ/FER WI) and physical mixing (CZZ/FER MIX).

On the whole, based on TEM characterization, CZZ/FER samples obtained by co-precipitation show the most promising structure of the catalyst in terms of inter-dispersion among the different phases (especially for the 2:1 loading): these findings are in accordance with results obtained from the characterization of the catalytic performance, as discussed in the other sections.

#### 3.1.2. Redox Properties and Acidity

Previous studies have proved that CuO and ZnO show a synergistic effect on the reducibility of both phases [81,82,83]. Really, a high dispersion of CuO and the presence of ZnO favors the reducibility of CuO at lower temperatures than “pure” CuO [83]. More specifically, the reduction peak of CuO has been usually deconvoluted in three peaks (see Appendix A) at increasing reduction temperature, corresponding respectively to highly dispersed CuO particles, and CuO particles in contact with ZnO and bulky CuO species without ZnO contact [55,84]. Nevertheless, other authors have related them to subsequent reduction steps of CuO to metallic Cu [55]. It is worth noting that the amount of H_2_ consumed during the reduction of the hybrid catalysts is proportional to the metallic loading; furthermore, the main reduction peak of CuO of each catalyst is located between 180 °C and 350 °C, thus a beneficial effect of ZnO on the reducibility of CuO is evident with respect to the reference CuO. Quantitative data reported in Table 1 point out that the specific H_2_ consumption of the main reduction peak (T_red_ < 350 °C) is directly proportional to the oxides/zeolite weight ratio. However, more interestingly is the extremely high experimental H_2_/CuO molar ratio (between 1.42 and 1.60) with respect to the theoretical stoichiometric H_2_/CuO ratio equal to 1.

The fraction of CuO reduced by H_2_ to metallic Cu was evaluated considering the H_2_ absorbed within the structure of ZnO. More specifically, H_2_-TPR profiles were fitted considering three peaks at low temperature (<350 °C) and a less resolved contribution (δ-peak) at high temperature (350–800 °C), which corresponds to the reduction of ZnO to metallic Zn. Some authors have reported that a synergistic effect between CuO and ZnO results in a reduction of CuO and a partial reduction of ZnO at a low temperature [81,83,85]. Nevertheless, this interpretation is in contrast with previous findings from XRD. Thus, it is more likely that nanoparticles of ZnO can store a huge amount of hydrogen within their crystalline structure [86,87,88,89] so that atomic hydrogen can occupy interstitial position on the surface and in the subsurface regions (<10 atomic layers) [86,87]. In addition, H_2_ adsorbs and dissociates on the surface of metallic Cu, and then atomic H migrates toward the oxide structure of ZnO [86]. Therefore, an intimate contact of Cu and ZnO particles emphasizes this phenomenon [87].

What stands out from Appendix A and Table 1 is that hybrid catalysts are more reducible than the physical mixed catalyst, the incomplete reduction of the last sample being in accordance with findings from XRD characterization (see Appendix A), where only diffraction peaks of metallic copper at low intensity were observed. Gel-oxalate coprecipitation appears as the best technique for producing high reducible copper particles with intimate contact with ZnO. Furthermore, the area of the α-peak of the H_2_-TPR profile decreases, while the peak temperature of the γ-peak rises as the average crystallite size of CuO increases; hence, α-peak is diagnostic of highly dispersed CuO particles, whereas the γ-peak accounts for the reduction of bulky CuO species.

As reported in Table 1, ZnO was completely reduced by H_2_ to metallic Zn at high temperatures (approximately 670 °C). Contrary to expectations, the peak temperature of ZnO reduction (δ-peak) decreases as both the ZnO and CuO average crystallite sizes increase. Yet, a signal drift below the baseline level suggests the release of interstitial hydrogen from the ZnO structure, which is no longer not stored within the oxide structure after the reduction of ZnO to metallic Zn.

XPS analyses was performed on calcined and spent samples to get information regarding the oxidation state of all the elements present. In Appendix A the relative atomic concentration (at. %) determined by high resolution spectra for all the calcined samples is reported.

As reported in Figure 4, the Cu2p doublet has been acquired for all the samples together with a metal reference, which has been previously Ar^+^ sputtered to remove oxide surface layer.

A clear feature due to Cu(II) oxidation state is the presence of a shake-up satellite, located between Cu2p_3/2_ and Cu2p_1/2_ peaks (and another paired satellite at higher binding energy after Cu2p_1/2_). This satellite peak, due to outgoing photoelectrons that interact with a valence electron and excite it to a higher-energy level, is not present either in Cu(0) (see metallic Cu reference curve in Figure 4 or in Cu(I)). An intensity decrease of this secondary signal is clear evidence that the reduction process has occurred [90]. Differently, after the catalytic run, a certain re-oxidation takes place, this evidence being in accordance with the XRD patterns. On the spent samples, a decrease in the Cu2p_3/2_ peak FWHM and a shift toward a lower binding energy with respect to the contribution of metallic copper (932.7 eV) are also observable.

In Appendix A, Zn2p, Zr3d, Si2p and O1s high resolution spectra were also compared, in order to check for other clear changes in the oxidation state of the selected elements. All the samples showed a main peak located at ~1022 eV, which is due to ZnO species [91]. Zr3d doublet also appear to be mostly identical for all the samples, and the peaks position has been ascribed to stoichiometric zirconia [92,93] without any contribution due to sub-oxide states. Additionally, Si2p peak has shown a main peak centered at 103.7 eV due to SiO_2_ chemical shift. At 101.6 eV the SiO contribution can also be recognized, while at 100.7 eV the Si_2_O signal is diagnostic of the Si2p peak position shifted at lower binding energy, as a typical feature of the CuZnZr-ferrierite based catalysts prompted by the synthesis environment [94]. For the O1s peak, all samples show three main signals (denoted as A, B, and C), which are attributed to the lattice oxygen (metal oxide), surface-adsorbed oxygen (organic compound) and surface hydroxyl species (metal carbonate or hydroxide), respectively [93]. It is not easy to separate the specific contribution of each species since there is an overlap among all of them. However, the spent catalysts exhibited a decrease in the signal intensity of the O1s contribution located at ~529.5 eV (peak A), as the result of a clear loss of lattice oxygen (oxides) caused by the Cu reduction from (II) to (0) during the catalyst activation or under the reducing atmosphere of the catalytic run.

Quantitative results of NH_3_-pulse chemisorption, summarized in Table 2, revealed a small variation in total number of acid sites between fresh and spent catalysts. Commercial bare FER is characterized by a concentration of acid sites of approximately 500 μmol_NH3_/g_cat_ [22,56], while fresh reduced hybrid catalysts obtained by gel oxalate coprecipitation and wet impregnation exhibit a greater concentration of acid sites with respect to the bare ferrierite. These results are in accordance with findings from other authors, considering that the metal deposition masks some acid sites of the zeolite [76,95]. Furthermore, the fraction of strong acid sites in the hybrid catalysts increased with respect to the bare ferrierite. According to FT-IR studies, this feature has been ascribed to an ion exchange phenomenon that occurs during the deposition of the metal load by forming Lewis sites [56,57,75,76,82].

On the other hand, the spent CZZ/FER OX 1:2 and CZZ/FER WI 1:2 catalysts exhibit a loss of acidity, so the blockage or the exchange of acid sites is likely to occur during activity tests [22,83]. Furthermore, the CZZ/FER MIX 1:2 sample displays an acid capacity (405 μmol_NH3_/g_FER_) comparable to the bare FER, evidently due to its preparation procedure. In fact, CuZnZr and FER powders were physically mixed to obtain the bifunctional catalyst, without any contribution to the total NH_3_ uptake brought by the metal-oxides at level of solid-state interaction.

### 3.2. Catalytic Screening in the Direct Hydrogenation of CO_2_ to DME

#### 3.2.1. Activity Tests

The catalytic screening for the direct CO_2_ hydrogenation to DME were conducted at 2.5 MPa, inlet H_2_/CO_2_/N_2_ molar ratio equal to 3:1:1, varying the reaction temperature in the range of 200–300 °C and the WHSV between 6.7 NLg_cat_^−1^∙h^−1^ and 20 NL∙g_cat_^−1^∙h^−1^ (catalyst load = 1.5 g).

As illustrated in Figure 5, irrespective of the preparation method, the activity of catalysts at an oxides/ferrierite weight ratio equal to 1:2 follows a similar trend, according to which the conversion of CO_2_ increases with the reaction temperature. In particular, CZZ/FER OX 1:2 appears to be the least active synthesized catalyst under the adopted operating conditions, exhibiting CO_2_ conversion values ranging from ~1% at 200 °C to ~14% at 300 °C. As expected, the performance of all the samples is very sensitive to the WHSV; in fact, the CO_2_ conversion at 275 °C doubles by tripling the residence time. On the contrary, the CZZ/FER WI 1:2 catalyst is less affected by a change of the WHSV, also attaining the same performances of the CZZ/FER OX 1:2 at a high temperature. In our previous work [22], a completely different result was obtained at 5.0 MPa and 8.8 NL∙g_cat_^−1^∙h^−1^, wherein CZZ/FER WI 1:2 revealed an extremely poor activity (from ~5% at 200 °C to ~14% at 260 °C) compared to CZZ/FER OX 1:2 (from ~9% at 200 °C to ~21% at 260 °C). Evidently, a reduction in partial pressure of the reactants mainly affects the activity of the CZZ/FER OX 1:2. Moreover, the CO_2_ conversion of the CZZ-FER MIX 1:2 catalyst is extremely low at 200 °C, while it rapidly increases by increasing the temperature to 18.5% at 300 °C and 6.7 NL∙g_cat_^−1^∙h^−1^.

CZZ/FER OX 1:2 and CZZ/FER WI 1:2 catalysts were less active than the mixed system, but their DME yield was extremely higher as the result of a lower activity in CO_2_ hydrogenation to CO through RWGS. In particular, the maximum DME yield was obtained at 275 °C when operating at 6.7 NL∙g_cat_^−1^∙h^−1^, reaching 2.1% (26.7 mg_DME_∙g_cat_^−1^∙h^−1^), 2.06% (26.4 mg_DME_∙g_cat_^−1^∙h^−1^) and 1.18% (15.3 mg_DME_∙g_cat_^−1^∙h^−1^) for CZZ/FER OX 1:2, CZZ/FER WI 1:2 and CZZ-FER MIX 1:2, respectively. As expected, the DME yield decreases at a higher WHSV, but the specific productivity of DME (mg_DME_∙g_cat_^−1^∙h^−1^) approximately doubles. By looking at the DME selectivity, a typical trend decreasing with the reaction temperature was recorded, with the maximum values of 53%, 47% and 33% obtained at 225 °C and 13.3 NL∙g_cat_^−1^∙h^−1^ for the CZZ/FER WI 1:2, CZZ/FER OX 1:2 and CZZ-FER MIX 1:2 samples, respectively. Furthermore, it is worth noting that the DME selectivity does not change sensibly by varying the residence time. As far as the CO yield is concerned, RWGS reaction starts to prevail at 250 °C since it becomes thermodynamically favored by increasing the reaction temperature. On this account, CZZ-FER MIX 1:2 was the most selective system toward CO, while CZZ/FER WI 1:2 exhibits the lowest CO selectivity in any experimental conditions. For the sake of completeness, it is worth mentioning that the uncertainty on the experimental data is generally greater at lower temperatures due to the low CO_2_ conversion and the smaller quantity of products at the outlet of the reactor.

Undergoing a fast dehydration rate, MeOH yield appears extremely low. However, it slightly increases as the temperature rises when the process approaches thermodynamic equilibrium, and the DME yield remains almost unchanged above 275 °C. On the whole, at an oxides/zeolite ratio of 1:2, the catalyst prepared by impregnation is less affected by an increase in WHSV, maintaining its highest DME yield even at a space velocity as high as 13.3 NL∙g_cat_^−1^∙h^−1^.

By increasing the oxides/zeolite weight ratio up to 2:1, what is striking is the completely different behavior observed in respect of the catalysts with a ratio of 1:2 (see Figure 6). CZZ/FER OX 2:1 is indeed extremely more active in CO_2_ hydrogenation, as the result of a double amount of metals. More specifically, the CO_2_ conversion of this sample is significantly affected by a variation in WHSV especially below 275 °C, approaching to the thermodynamic equilibrium at 300 °C. This aspect is easily explained by considering that CO_2_ is converted both to MeOH and CO on the metal-oxide sites of the CuZnZr phase, while the dehydration of MeOH into DME occurs on the acid sites of the ferrierite [22,40,42,96]. From a quantitative point of view, the overall specific surface area of CZZ/FER OX 2:1 is roughly halved with respect to the bare ferrierite, because oxides typically account for a lower surface area than zeolites. For these reasons, a larger amount of metal-oxide phase increases the site exposure for converting CO_2_ to MeOH and/or CO, while remaining adequate the accessible surface of the ferrierite to dehydrate MeOH to DME.

Really, taking advantage of its higher oxides/ferrierite weight ratio, the CZZ/FER OX 2:1 sample exhibited the best performance during direct hydrogenation of CO_2_ to DME. More specifically, coprecipitated CZZ/FER OX 2:1 exhibited a DME yield that peaked close to 5.0% (58.9 mg_DME_∙g_cat_^−1^∙h^−1^) at 250 °C and 6.7 NL∙g_cat_^−1^∙h^−1^, while remaining below 4% (138.3 mg_DME_∙g_cat_^−1^∙h^−1^) at 275 °C and 20.0 NL∙g_cat_^−1^∙h^−1^. In terms of MeOH yield, the highest value was obtained at 275 °C, roughly attaining 1.4%, whereas due to a larger number of metallic sites on this hybrid catalyst, the CO yield approximately follows the thermodynamic profile because of a large contribution of RWGS driven by a metal-based phase. For the sake of clarity, in Figure 5b and Figure 6b, the experimental CO selectivity is greater than the thermodynamic at low temperatures because the average reaction rate of CO production at the reaction conditions is relatively higher than the reaction rate of CO production at the equilibrium.

#### 3.2.2. Structure–Activity Relationships

The results obtained from the experimental tests of catalytic activity are certainly to be related to the physico-chemical properties of the catalytic materials. It is worth pointing out that the behavior and characteristics of the metal oxides and of the zeolite are completely different from each other; therefore, catalytic performances are the result of a set of distinctive characteristics led by the different preparation procedures and interactions between metal-oxides and zeolite phase.

As seen in Appendix A, the total specific surface area and the specific micropores area decrease as the metallic load in the catalyst increases, while the average pore diameter increases. Considering that, as shown in Figure 7a, the specific CO_2_ conversion rate decreases as the total specific surface area rises, it is evident that increasing the metallic sites by increasing the mass of CZZ, reduces the total surface area of the catalyst. The oxides have indeed a lower specific surface area than the ferrierite; moreover, the deposition (impregnation or co-precipitation), may occlude pores of the zeolite. Hence, an increase of the metal oxide/zeolite mass ratio favors the methanol synthesis, but the mass of zeolite decreases (by keeping constant the overall mass of catalyst) and the limiting reaction step from the methanol synthesis becomes the methanol dehydration and so the overall CO_2_ conversion becomes independent of the increase of the oxide/zeolite mass ratio. Furthermore, this effect appears more pronounced at a higher temperature, wherein the formation of CO becomes preferential, hindering the consecutive dehydration path of methanol to DME, in turn limiting the driving force for CO_2_ conversion. According to the literature, the oxide/zeolite mass ratio exhibits a maximum that represents the best compromise between the methanol synthesis that occurs on the metallic phase and the methanol dehydration that occurs on the acid sites of the zeolite [25]. In this work, the catalyst with an oxide/zeolite ratio equal to 2 (i.e., CZZ/FER OX 2:1) seems to exhibit the best performances.

In addition, as shown in Figure 7b, not only the metal loading but also the size of the metallic copper crystallites significantly affects the specific CO_2_ conversion rate. In particular, a volcano-shaped trend is observed at any temperature, according to which the CO_2_ conversion rate goes through a maximum for intermediate sizes of metallic Cu crystallites (i.e., ~17 nm). Since the dispersion of the metals decreases as the size of the Cu crystallites increases, consequently the number of active sites for the conversion of CO_2_ also diminishes. Although a smaller size of the metallic Cu crystallites accounts for a higher dispersion of metals and number of active sites, the small crystallites tend to aggregate thus forming metal clusters at a low extent of metal-oxide interface, basically representing the surface region where CO_2_ is more likely activated [22,25,56,62]. Likewise, very large crystallites (>20 nm) also depress the catalytic activity, as the result of a poor metal–oxide interaction exhibited by the mixed CZZ-FER MIX 1:2 sample or macroscopic segregation of oxides during impregnation of ferrierite in CZZ/FER WI 1:2, well evidenced in FESEM images and EDS mapping. Obviously, the extent of the metal–oxide interaction also depends on the dispersion of ZnO, which is thermally more stable with smaller crystallite sizes (~8 nm), enhancing the conversion of CO_2_ [86].

A higher initial crystallite size of CuO is also put in relation both with a reduction of the α-peak of the H_2_-TPR profile and an increase in the area of the γ-peak of the H_2_-TPR, being that the bulk CuO of crystallites and nanoparticles is more difficult to be reduced. Similarly, the H_2_ consumption of the low temperature peak (i.e., α + β + γ) decreases as the size of the ZnO crystallites increases since the crystallites are larger and the atomic H can also diffuse within the ZnO structure [86,88,89]. At the same time, the fraction of copper and zinc on the surface of the material measured by using XPS increases with the average size of the copper and ZnO crystallites. A greater quantity of ZnO exposed on the catalyst surface seems to improve the rate of CO_2_ conversion, with a direct effect on DME selectivity at expenses of CO. On the other side, at an oxides/weight ratio as high as 2:1, a decrease in the surface atomic concentration of SiO_2_ leads to a decrease in the total surface acidity of the samples, directly affecting the methanol dehydration step to DME and conversely favoring an increase in CO selectivity.

#### 3.2.3. Stability Tests

In Figure 8, the results of the stability tests on the CuZnZr ferrierite-based hybrid catalysts are reported. They were carried out at 13.3 NL∙g_cat_^−1^∙h^−1^ and 275 °C for ~20 h in order to stress the catalysts in a comparable range of performances.

The CO_2_ conversion profile of each durability test was then fitted by using an exponential function, as reported in Equation (6) [56]; where $\zeta_{CO_{2},0}$ and $\zeta_{CO_{2}}\left(t\right)$ represent, respectively, the initial CO_2_ conversion and the CO_2_ conversion at the t-th time, k_d_ (h^−1^) is the deactivation constant and t (h) is the time.
(6)$$\zeta_{CO_{2}}\left(t\right)=\zeta_{CO_{2},0}\cdot \exp\left(-k_{d}\cdot t\right)$$

As shown in Figure 8, irrespective of the oxides/ferrierite weight ratio, the CO_2_ conversion trend is quite similar for the samples prepared by gel-oxalate coprecipitation and impregnation, showing a comparable deactivation trend, despite the difference in the initial activity. Instead, the mixed CZZ-FER MIX 1:2 system exhibits a flatter pattern without any visible loss of activity after 20 h with respect to the fresh catalyst. As previously discussed, such an almost constant stability pattern of the CZZ/FER MIX 1:2 sample is mainly to be related to a low mutual interaction among sites of different nature, prompted by the mixing of the two preformed catalysts (i.e., CuZnZr and FER) which, on one hand, brings to low activity but, on the other hand, to high stability. Whereas, on the other samples a higher synergy among metal-oxide-acid sites during the synthesis procedure, as highlighted by microscope investigations, negatively affects the evolution of the activity during time-on-stream.

Appendix A summarizes the deactivation parameters of the exponential deactivation function of all hybrid catalysts; in more detail, CZZ/FER WI 1:2 exhibited the highest loss in activity (9.36∙10^−3^ h^−1^), whereas the physically mixed catalyst was the most stable (1.17∙10^−3^ h^−1^). The loss in activity and the change in selectivity can really be associated to several phenomena. According to the open literature, the water produced during the process could inhibit active sites of the catalyst and mainly affect the acid sites of the zeolite [40,56,97]. The catalytic performance during a duration test is indeed sensible to the variation of spatial velocity, which can determine important physico-chemical modifications depending on its extent [56]. As reported by Bonura et al. [56], the transitional behavior of the selectivity profiles of DME, MeOH and CO lasts ~3 h at 13.3 NL∙g_cat_^−1^∙h^−1^, so that after the initial modifications, the selectivity profiles of each catalyst are roughly constant. In this initial loss in activity a certain role is also exerted by metal sintering, favored by reaction conditions as well as by location and availability of the acid sites more or less prone to adsorb water formed during run [56,81,86]. As illustrated in Figure 9, the deactivation rate increases with the acidity of the samples; this trend being more evident when an intimate contact between the metal-oxide phase and the zeolite is realized [56].

Furthermore, such a close synergy among sites can also cause a fast deactivation of the hybrid catalysts, due to a possible migration of metal clusters induced by water formed during reaction and promoting an exchange with the acid sites of the zeolite [81,85]. Another deactivation phenomenon may be coking; however, Miletto et al. [75] and Bonura et al. [56] have demonstrated the absence of carbon deposition on the surface of spent CuZnZr-ferrierite catalysts. Therefore, this deactivation mechanism was excluded. Other studies have reported a progressive deactivation of hybrid catalysts, following a significant loss in CO_2_ conversion and DME selectivity at expense of an increase in CO selectivity during stability tests at high residence time [40,56]. Nonetheless, as shown in Appendix A, despite a progressive loss in activity, the stability tests displayed only slight changes at the level of selectivity patterns. Further investigations should be carried out to better investigate the long-term modifications that occur in the catalysts by performing durability tests of at least 100 h.

## 4. Conclusions

In this work, the behavior of differently prepared CuZnZr ferrierite-based hybrid catalysts was investigated in light of physico-chemical modifications occurring during the direct hydrogenation of carbon dioxide into dimethyl ether. The N_2_ physisorption and the porosimetry analyses highlighted a decrease in the surface area and a broadening of the mesopore size distribution in the spent catalysts, mainly related to the sintering of the metallic phase on the surface of the ferrierite, enhanced by a blockage of the active sites caused by wetting. Indeed, XRD analysis revealed a widening of the copper and zinc crystallites, whereas Zr was present in an amorphous phase, and along with XPS and H_2_-TPR characterizations an incomplete reduction of the CuO below 350 °C was also evidenced. Regarding the morphological analysis performed by FESEM, EDS and TEM on the synthesized catalysts, it was observed how the distribution of the oxide phase on the surface of the zeolite significantly affects the mass-transfer phenomena among neighboring sites within the catalyst.

During the catalytic tests, the CZZ/FER OX 2:1 hybrid catalyst, prepared by gel-oxalate coprecipitation at an oxides/zeolite weight ratio as high as 2:1, reached a CO_2_ conversion close to 5.0% (58.9 mg_DME_∙g_cat_^−1^∙h^−1^) at 250 °C and 6.7 NL∙g_cat_^−1^∙h^−1^. As for the samples with an oxides/zeolite weight ratio of 1:2, the highest activity was obtained on the CZZ/FER WI 1:2 and the CZZ/FER OX 1:2 hybrid catalysts, with a comparable DME yield of ~2.0% (20–30 mg_DME_∙g_cat_^−1^∙h^−1^) at 275 °C and 6.7 NL∙g_cat_^−1^∙h^−1^. Furthermore, their activity decreased as the WHSV increased and the CZZ/FER OX 1:2 seemed to be more sensible to a variation in the residence time.

The stability tests showed a progressive decrease in the CO_2_ conversion, as the result of a mix of phenomena prompted by water formation under the adopted experimental conditions and ascribable to the metal sintering of the active phase, a change in the development of the surface area and/or a loss of surface acidity for possible exchange of metal atoms (or clusters) with acid sites of the zeolite. These phenomena appeared less marked on the sample in which the preformed methanol synthesis catalyst was barely mixed with the ferrierite, even if the random and limited synergy among the sites generated behind this simple preparation procedure determined a poorer performance than the hybrid catalysts, instead characterized by a mutual cooperation of sites suitable to drive the hydrogenation of CO_2_ directly to DME.

## Figures and Tables

**Figure 1 materials-15-07774-f001:**
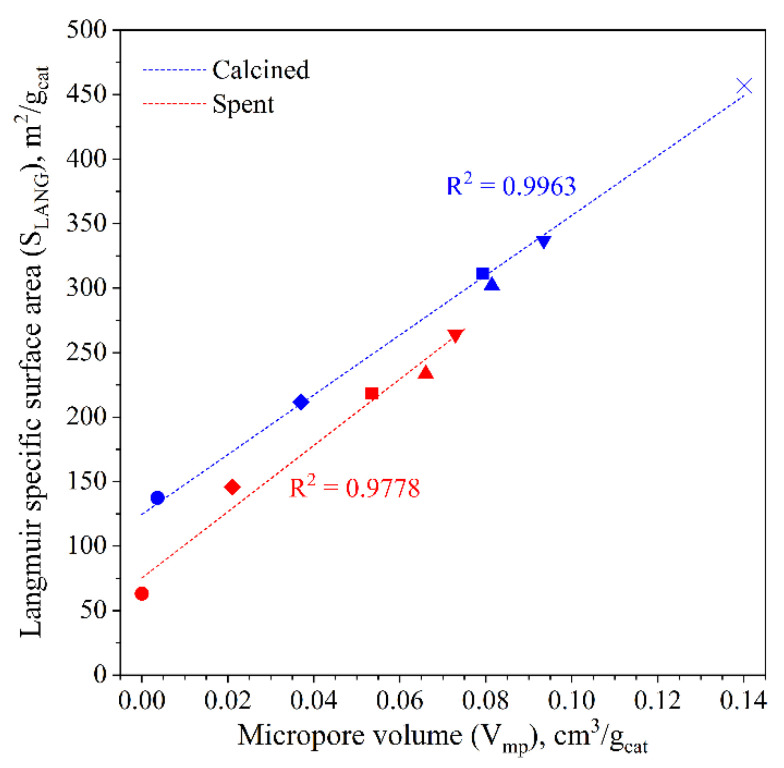
Relationship between the micropore volume and the specific surface area for calcined and spent catalysts (● CuZnZr; ♦ CZZ/FER OX 2:1; ■ CZZ/FER OX 1:2; ▲ CZZ/FER WI 1:2; ▼ CZZ-FER MIX 1:2; ×Ferrierite).

**Figure 2 materials-15-07774-f002:**
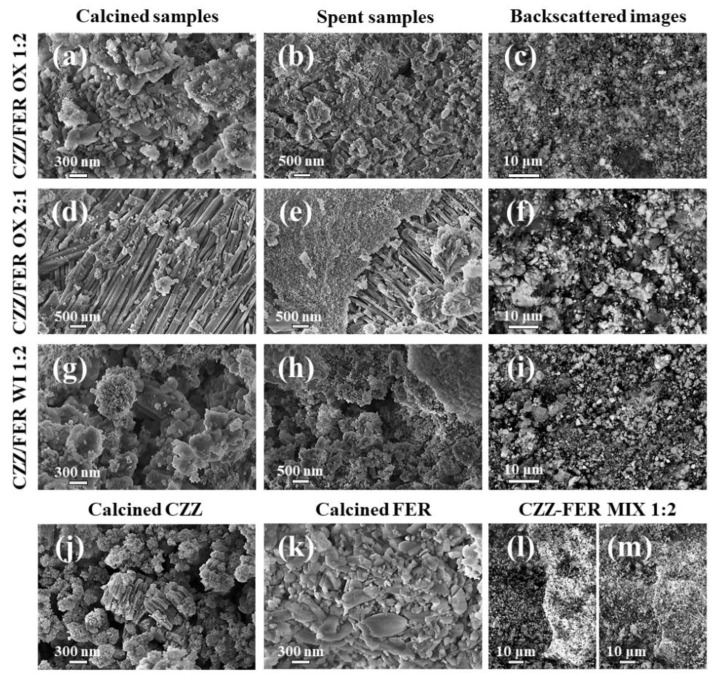
FESEM images of calcined samples: (**a**) CZZ/FER OX 1:2, (**d**) CZZ/FER OX 2:1, (**g**) CZZ/FER WI 1:2, (**j**) CZZ, (**k**) commercial FER and (**m**) CZZ-FER MIX 1:2. FESEM images of spent catalysts: (**b**) CZZ/FER OX 1:2, (**e**) CZZ/FER OX 2:1 and (**h**) CZZ/FER WI 1:2. Backscattered FESEM images of calcined samples: (**c**) CZZ/FER OX 1:2, (**f**) CZZ/FER OX 2:1, (**i**) CZZ/FER WI 1:2 and (**l**) CZZ-FER MIX 1:2.

**Figure 3 materials-15-07774-f003:**
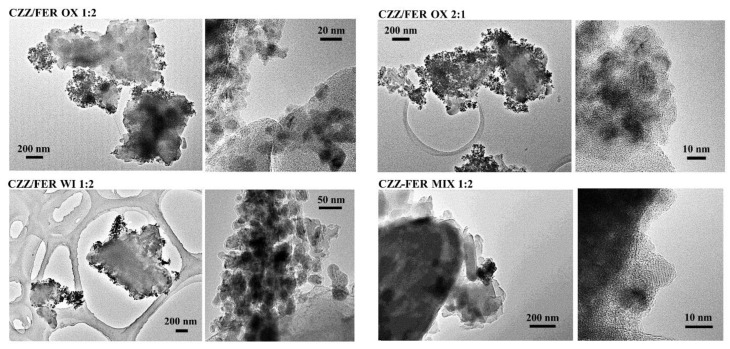
Bright-field TEM images of spent hybrid catalysts.

**Figure 4 materials-15-07774-f004:**
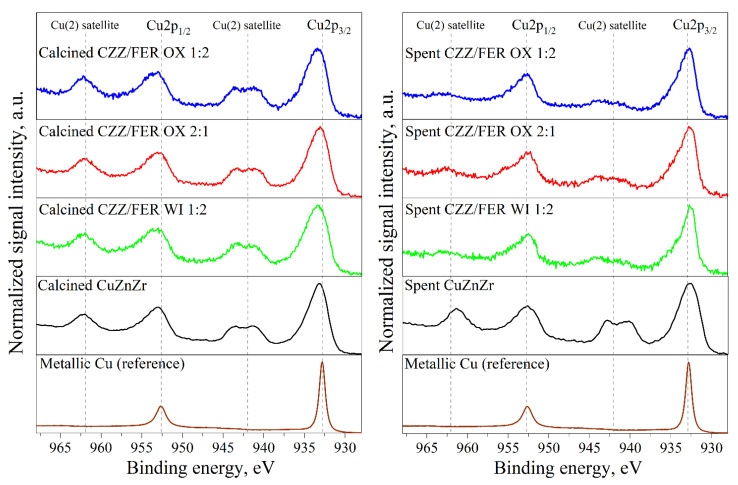
Cu2p high-resolution spectra for all the XPS tested samples and metallic Cu reference for comparison. For clearness, each spectrum was normalized with respect to the maximum height of the Cu2p_3/2_ peak.

**Figure 5 materials-15-07774-f005:**
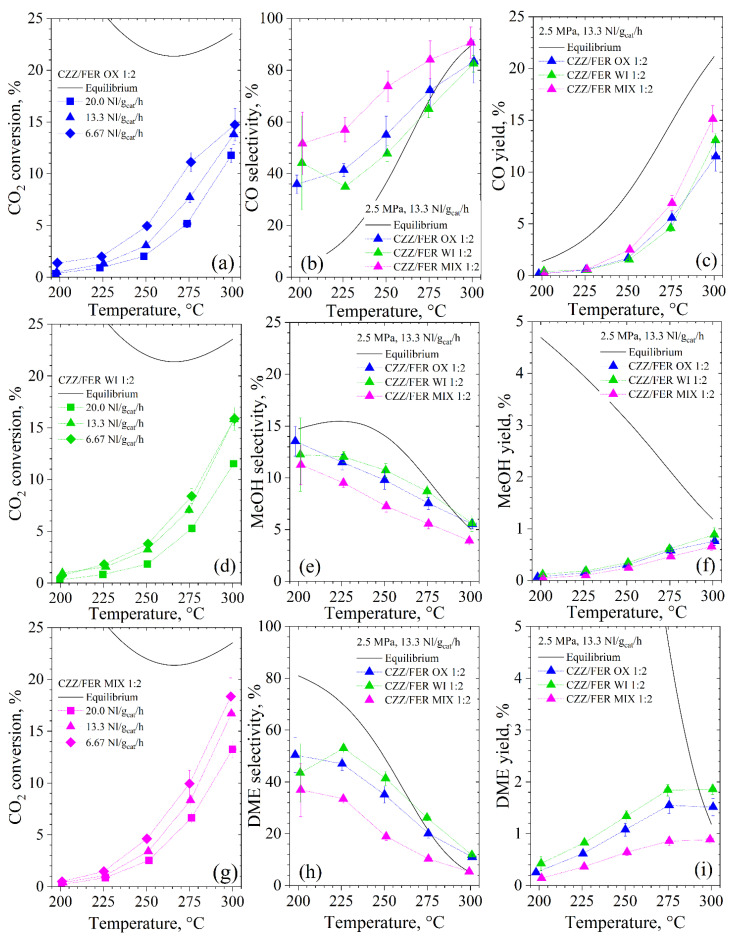
CO_2_ conversion of (**a**) CZZ/FER OX 1:2, (**d**) CZZ/FER WI 1:2 and (**g**) CZZ-FER MIX 1:2. (**b**) CO, (**e**) CH_3_OH and (**h**) DME selectivity at 13.3 NL∙g_cat_^−1^∙h^−1^ and (**c**) CO, (**f**) CH_3_OH and (**i**) DME yield at 13.3 NL∙g_cat_^−1^∙h^−1^. Reaction conditions: 2.5 MPa, inlet molar ratio H_2_/CO_2_/N_2_, 3/1/1.

**Figure 6 materials-15-07774-f006:**
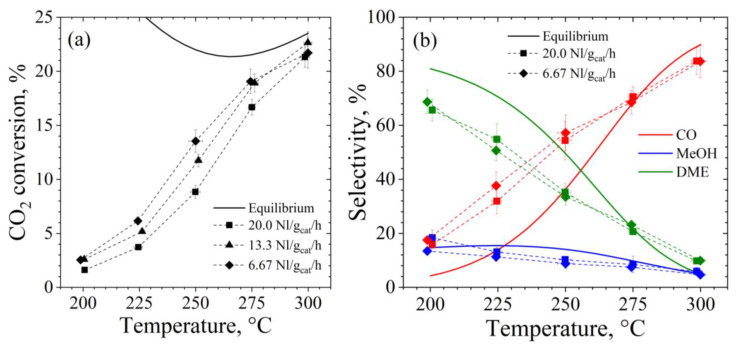
(**a**) CO_2_ conversion and (**b**) CO, CH_3_OH and DME selectivity of CZZ/FER OX 2:1. Reaction conditions: 2.5 MPa, inlet molar ratio H_2_/CO_2_/N_2_, 3/1/1.

**Figure 7 materials-15-07774-f007:**
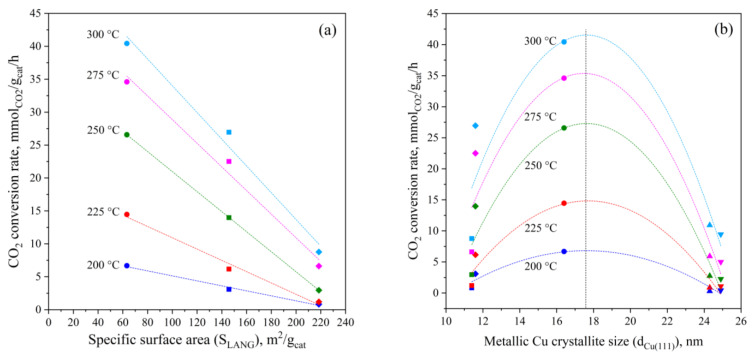
Relationships between the CO_2_ conversion rate and (**a**) the specific surface area and (**b**) the average crystallite size of the metallic Cu of the spent samples. Reaction conditions: 2.5 MPa; inlet H_2_/CO_2_/N_2_, 3/1/1 mol/mol; WHSV, 20 NL∙g_CuZnZr_^−1^∙h^−1^ (● CuZnZr; ♦ CZZ/FER OX 2:1; ■ CZZ/FER OX 1:2; ▲ CZZ/FER WI 1:2; ▼ CZZ-FER MIX 1:2).

**Figure 8 materials-15-07774-f008:**
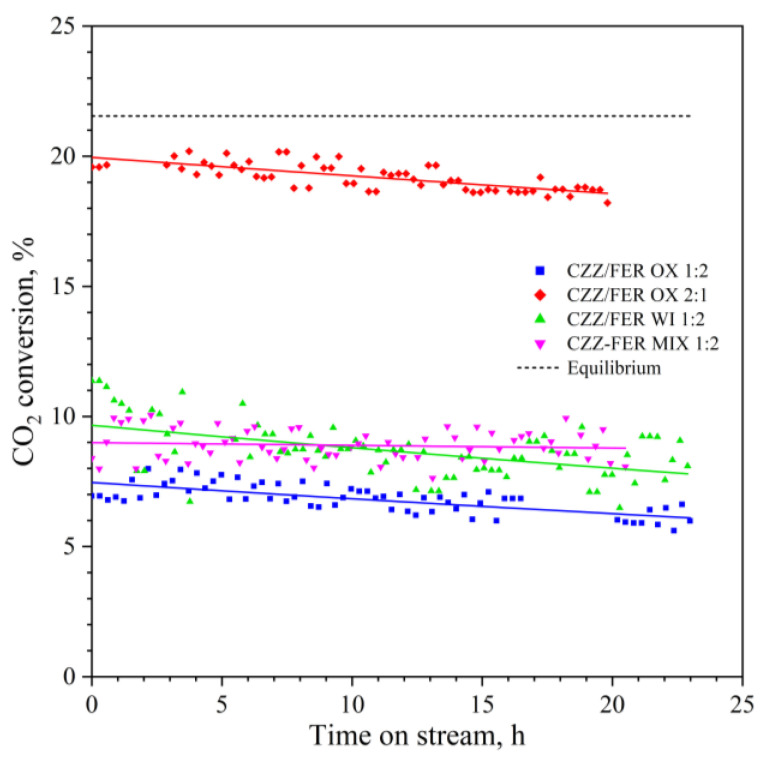
CO_2_ conversion during stability tests (~20 h). Reaction conditions: 2.5 MPa; 275 °C; inlet H_2_/CO_2_/N_2_, 3/1/1 mol/mol; WHSV, 13.3 NL∙g_cat_^−1^∙h^−1^. The solid lines represent the interpolated exponential function of the catalyst deactivation.

**Figure 9 materials-15-07774-f009:**
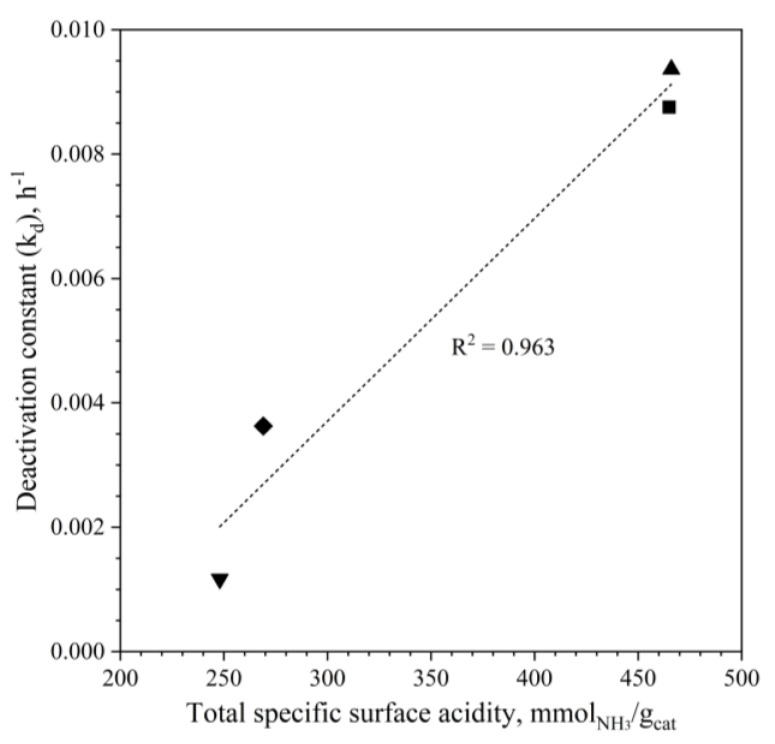
Relationship between the deactivation constant and the total specific surface acidity of the CuZnZr ferrierite-based hybrid catalysts (♦ CZZ/FER OX 2:1, ■ CZZ/FER OX 1:2, ▲ CZZ/FER WI 1:2 and ▼ CZZ-FER MIX 1:2).

**Table 1 materials-15-07774-t001:** Quantitative analysis of the H_2_-TPR measurements.

Sample	H_2_ Consumption (mmol_H2_/g_cat_)		H_2_/CuO ^1^	Reduced CuO ^2^ (%)	Reduced ZnO ^3^ (%)	Reduction Peaks ^4^ (%)		
	T_red_ ≤ 350°C	δ Peak				α	β	γ
CuZnZr	0.94	0.28	0.13	13	8	10	46	44
CZZ/FER OX 2:1	6.83	2.32	1.442	95	98	19	81	-
CZZ/FER OX 1:2	3.67	1.18	1.42	~100	~100	24	62	14
CZZ/FER WI 1:2	3.80	1.18	1.60	94	~100	8	81	11
CZZ-FER MIX 1:2	1.95	0.51	0.82	82	7	12	13	57

^1^ The H_2_/CuO molar ratio was evaluated considering the area of the reduction peaks at T ≤ 350 °C and the nominal composition of the catalyst. ^2^ The fraction of reduced CuO to metallic Cu at T ≤ 350 °C was evaluated subtracting the desorbed amount of H_2_ at T > 350 °C to the area of the reduction peak at T ≤ 350 °C considering the nominal composition of the catalyst. ^3^ The fraction of reduced ZnO to metallic Zn at T > 350 °C was evaluated using the nominal composition of the catalysts. ^4^ The fractions of α, β and γ peaks were evaluated as the ratio between the area of a deconvoluted peak and their total area.

**Table 2 materials-15-07774-t002:** Quantitative data of NH_3_-pulse chemisorption.

Catalysts	State	NH_3_ Uptake		Weak Acid Sites ^1^	Strong Acid Sites ^2^
		μmol_NH3_/g_cat_	μmol_NH3_/g_FER_	%	%
Commercial FER	Fresh reduced	494 ± 50	494 ± 50	80	20
	Spent	-	-	-	-
CZZ/FER OX 1:2	Fresh reduced	540 ± 37	810 ± 56	35	65
	Spent	465 ± 12	698 ± 18	41	59
CZZ/FER WI 1:2	Fresh reduced	511 ± 24	767 ± 36	35	65
	Spent	466 ± 17	699 ± 26	35	65
CZZ-FER MIX 1:2	Fresh reduced	270 ± 31	405 ± 47	60	40
	Spent	248 ± 25	372 ± 38	64	36
CZZ/FER OX 2:1	Fresh reduced	271 ± 15	813 ± 45	42	58
	Spent	269 ± 34	807 ± 102	28	72

^1^ It represents the fraction of NH_3_ desorbed below 300 °C. ^2^ It represents the fraction of NH_3_ desorbed above 300 °C.

## Data Availability

Data sharing not applicable.

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
