# Peer review of "Physico-Chemical Modifications Affecting the Activity and Stability of Cu-Based Hybrid Catalysts during the Direct Hydrogenation of Carbon Dioxide into Dimethyl-Ether"

_materials, 2022, doi:10.3390/ma15217774_

Round 1

Reviewer 1 Report

The authors synthesized one ferrierite-based CuZnZr hybrid catalysts that can directly hydrogenate CO2 into dimethyl-ether (DME) by carrying out in a fixed bed reactor. Experimental results show that the catalyst can realize a maximum DME yield as high as 4.5 % (58.9 mgDME∙gcat-1∙h-1). The established mesoporous metal catalysts are meaningful for the hydrogenation of CO2 with high catalytic activity and selectivity. Therefore, I would like to recommend its acceptance by the journal after a minor revision as follow.

1, Why the physically mixed sample also showed a high activity in CO2 hydrogenation?

2, Temperature is vital for the hydrogenation of CO2 into DME. Can be the reaction carried out under much lower temperature (such as the ambient temperature)?

3, How to fit and obtain the curve in Figure 7b? I doubt that it is overfitted.

4, The recent studies on the modifications of hybrid catalysts should be cited. For example, Exploration 2022, 2, 20210095; CCS Chem. 2022, Just Published. DOI: 10.31635/ccschem.022.202202328. 

Reviewer 2 Report

-equation 1 should be corrected (2CO2 + 6H2O --> CH3OCH3 + 3H2O).
-the effect of oxalic should be addressed in the introduction. 
-In fig. 5 c, the color of CZZ/FER MIX 1:2 should be purple, and CZZ/FER WI 1:2 should be green.
-In fig. 5 b, please give more detail why CO selectivity if CZZ/FER WI 1:2 at 200oC was not the same trend as other catalysts? 

-in stability test, water and metal sintering were mentioned as main cause of deactivation, how about the coking?
